# Anti-Inflammatory and Antiviral Effects of Cannabinoids in Inhibiting and Preventing SARS-CoV-2 Infection

**DOI:** 10.3390/ijms23084170

**Published:** 2022-04-10

**Authors:** Marcin Janecki, Michał Graczyk, Agata Anna Lewandowska, Łukasz Pawlak

**Affiliations:** 1Department of Palliative Care and Palliative Medicine, Silesian Medical University in Katowice, 40-752 Katowice, Poland; mjanecki@sum.edu.pl; 2Department of Palliative Care, Collegium Medicum in Bydgoszcz, Nicolaus Copernicus University in Toruń, 87-100 Toruń, Poland; 310th Military Research Hospital and Polyclinic in Bydgoszcz, 85-681 Bydgoszcz, Poland; 4Collegium Medicum in Bydgoszcz, Nicolaus Copernicus University in Toruń, 87-100 Toruń, Poland; lukasz.pawlak.lpp@gmail.com

**Keywords:** *Cannabis sativa*, cannabinoids, SARS-CoV-2, COVID-19, anti-inflammatory, antiviral, CBD, terpene

## Abstract

The COVID-19 pandemic caused by the SARS-CoV-2 virus made it necessary to search for new options for both causal treatment and mitigation of its symptoms. Scientists and researchers around the world are constantly looking for the best therapeutic options. These difficult circumstances have also spurred the re-examination of the potential of natural substances contained in *Cannabis sativa* L. Cannabinoids, apart from CB1 and CB2 receptors, may act multifacetedly through a number of other receptors, such as the GPR55, TRPV1, PPARs, 5-HT1A, adenosine and glycine receptors. The complex anti-inflammatory and antiviral effects of cannabinoids have been confirmed by interactions with various signaling pathways. Considering the fact that the SARS-CoV-2 virus causes excessive immune response and triggers an inflammatory cascade, and that cannabinoids have the ability to regulate these processes, it can be assumed that they have potential to be used in the treatment of COVID-19. During the pandemic, there were many publications on the subject of COVID-19, which indicate the potential impact of cannabinoids not only on the course of the disease, but also their role in prevention. It is worth noting that the anti-inflammatory and antiviral potential are shown not only by well-known cannabinoids, such as cannabidiol (CBD), but also secondary cannabinoids, such as cannabigerolic acid (CBGA) and terpenes, emphasizing the role of all of the plant’s compounds and the entourage effect. This article presents a narrative review of the current knowledge in this area available in the PubMed, Scopus and Web of Science medical databases.

## 1. Introduction

The endocannabinoid system (ECS) is composed of two key receptors, named cannabinoid type 1 (CB1) and cannabinoid type 2 (CB2), endogenous cannabinoids binding the receptors, such as anandamide (AEA) and 2-arachidonoylglycerol (2-AG), together with the enzymes responsible for their synthesis and degeneration, such as fatty acid amide hydrolase (FAAH) for AEA and monoacylglycerol lipase (MAGL) for 2-AG [1,2,3]. The CB1 receptor is one of the G protein-coupled receptors located mainly in the central nervous system (CNS)—particularly abundantly in the neocortex, hippocampus, basal ganglia, cerebellum and brainstem [4]. It can also be found in peripheral organs, such as the liver, muscles and pancreas, and in adipose tissue, where it regulates food intake, energy expenditure and reward-related responses [1]. The CB2 receptor is highly expressed in immune cells, such as lymphocytes B, lymphocytes CD8, NK cells, monocytes, lymphocytes CD4 and neutrophils [5,6]. CB2 receptors present an anti-inflammatory effect by reducing the release of proinflammatory cytokines, increasing production of anti-inflammatory cytokines, inhibiting macrophage migration and regulating lymphocyte T activation [5].

Apart from binding to CB1 and CB2 receptors, cannabinoids alternatively affect numerous other receptors, such as G protein-coupled receptor 55 (GPR55), transient receptor potential vanilloid channels (TRPV) and a class of peroxisome proliferator-activated receptors (PPARs) [3,7]. Moreover, they exhibit affinity to the serotonin 1A receptor (5-HT1A), as well as adenosine and glycine receptors [7]. Multifaceted paths of action of cannabinoids involve their anti-inflammatory and antioxidative effects, together with the capacity to modulate immunological processes (Figure 1).

Endocannabinoid receptors are also activated by exogenous substances, known as phytocannabinoids—extracted from *Cannabis sativa* L., and other synthetic compounds [8]. *Cannabis sativa* L. is rich in over 100 discovered phytocannabinoids. The key two include Δ9-tetrahydrocannabinol (THC), a compound with psychoactive properties due to its strong affinity to CB1 receptors, and cannabidiol (CBD), which is considered non-psychotropic [8]. Other compounds of *Cannabis sativa* L. include cannabigerol (CBG), cannabichromene (CBC), tetrahydrocannabivarin (THCV), cannabicitran (CBT) and many others [8,9]. Moreover, the extract contains non-psychoactive essential oils (terpenes), which exhibit synergism in interactions with cannabinoids and enhance their medicinal functionality, which is known as the entourage effect [10].

THC binds primarily to CB1 receptors due to its lipophilic character and the ability to penetrate the blood–brain barrier, though there is a weak binding of CB2 receptors as well [11,12]. CBD has a low affinity to CB1 and CB2 receptors [13]. It is also a partial agonist of serotonin 5-HT1A receptor, TRPV1 receptor, an allosteric modulator of δ and μ opioid receptors, as well as PPAR-γ agonist and intracellular calcium release modulator [13,14]. As aforementioned, CBD does not have psychoactive properties, although the affinity to various receptors and signalling paths confirmed its anti-inflammatory and antioxidant effects [3]. Numerous diseases, such as anorexia, chronic pain, inflammation, multiple sclerosis, neurodegenerative disorders, epilepsy, cardiovascular disorders, cancers and metabolic syndrome-related disorders, to name just a few, are treated or have strong potential to be treated by cannabinoids or cannabinoid-related compounds [15].

The COVID-19 pandemic is caused by severe acute respiratory syndrome coronavirus 2 (SARS-CoV-2) [16,17]. The symptoms of COVID-19 can be divided into basic categories, such as flu-like symptoms (fever, fatigue, muscle aches, headaches), respiratory symptoms (cough, shortness of breath) and gastrointestinal symptoms (diarrhoea, nausea). Frequently patients experience anosmia and ageusia, which mean the loss of smell and taste, respectively, suggesting the neuroinvasive potential of the virus [16,17]. The clinical presentation of SARS-CoV-2 infection ranges from asymptomatic infection to severe pneumonia in the course of a cytokine storm and hyperinflammation, which ultimately lead to acute respiratory distress syndrome (ARDS), multiple organ failure and death [18]. People with a severe course of COVID-19 most often have at least one comorbid disease, such as hypertension, diabetes or cardiovascular diseases.

As the pathogenesis and complications of SARS-CoV-2 infection involve an immune-inflammatory cascade, treatment strategies mainly rely on immune modulation and a reduction in inflammation [6]. Considering the fact that SARS-CoV-2 infection leads to excessive immunological and inflammatory response, and that cannabinoids have the potential to regulate these processes, it can be assumed that they could be used in the treatment of COVID-19.

The SARS-CoV-2 virus was isolated in November 2019 and we systematically learn about the symptoms of the disease accompanying its variants. At that time, many publications on this subject appeared, including the potential influence of cannabinoids on the course of the disease caused by the SARS-CoV-2 virus. This article presents a narrative review of the current knowledge in this area. 

## 2. Antiviral Properties of Cannabinoids

SARS-CoV-2 is an RNA virus composed of a lipid bilayer and four structural proteins. The virus membrane consists of three integral proteins—the spike (S), the membrane (M) and the envelope (E) [19]. SARS-CoV-2 enters human cells by binding the S protein to the angiotensin converting enzyme 2 (ACE2) receptor, which is mainly expressed in lung tissue, as well as the oral and nasal mucosa, kidneys, testes and the gastrointestinal tract [17].

Excessive expression of ACE2 receptors is correlated to cytokine secretion and inflammatory response in COVID-19 patients [20]. The binding of the S protein and the ACE2 receptor leads to proteolysis of the S protein by transmembrane serine protease 2 (TMPRSS2) into two bound peptides—S1 and S2—which facilitate the viral entry into the cell by both binding to the ACE2 receptor and mediating membrane fusion [19]. In the host cell, the viral genome undergoes translation into two polypeptides, which later become cleaved by SARS-CoV-2 proteases—main (M^pro^) and papain-like (PL^pro^). The proteins produced by this process facilitate viral replication and enable the virus to spread inside the host organism [19].

The primary aim of the treatment used in COVID-19, in addition to reducing inflammation and symptoms associated with the excessive activation of the immune system, is to stop the virus from replicating in the organism. Natural compounds are promising targets for drug development, and a number of therapies have been inspired and based on natural products [21]. Molecular docking analysis revealed strong binding affinities between the phytocannabinoids and codon mRNAs of proteins implicated in replication, translation and release of SARS-CoV-2, which portrays cannabinoids as potential drugs against COVID-19 [22]. Moreover, inhibition of the main protease among coronaviruses—M^pro^, which plays a significant role in the COVID-19 virus replication process—is also considered a potential therapeutic target [23,24]. The search for substances (phytochemicals) with antiviral properties that could inhibit SARS-CoV-2 M^pro^ is still undergoing [23].

An in vitro study proved that phytocannabinoids—CBD and THC—have the ability to interact with SARS-CoV-2 M^pro^ as antagonists, leading to the inhibition of the virus translation and blocking its replication [5]. There are also reports of CBD’s ability to inhibit TMPRSS2 in several models of human epithelia [17,25]. Additionally, CBD and THC, acting as CB2 receptor agonists, reduce the level of pro-inflammatory cytokines in lung cells [5].

ACE2 receptors play a key role in the entry of the SARS-CoV-2 virus into the host cells. Downregulation of their expression in tissues may prove to be an important strategy in reducing the susceptibility to infection [17]. An in vitro study showed that high-CBD extracts from cannabis have the ability to downregulate the expression of ACE2 receptors in artificially inflamed tissues subjected to TNFα/IFN-γ [17]. The use of CBD alone did not reduce the expression of the ACE2 receptor. The results suggest a possibility that it is the entourage effect of the components of the extracts other than CBD that is responsible for the downregulation of ACE2 receptor expression [26]. Interestingly, the authors of another in vitro study achieved the reduction of ACE2 receptor expression in epithelial cells of lung tissue after administration of CBD [27].

Other phytocannabinoids—cannabigerolic acid (CBGA) and cannabidiolic acid (CBDA)—also prevented the entry of SARS-CoV-2 into human epithelial cells, as well as averted the infection induced by a pseudovirus expressing the SARS-CoV-2 spike protein. CBGA and CBDA were similarly effective against both alpha and beta variants of SARS-CoV-2 [28].

Another in vitro study determined that from the tested cannabis compounds, only CBD has antiviral properties and the ability to inhibit viral replication in SARS-CoV-2 infection. Moreover, the use of CBD in combination with THC significantly weakened the effectiveness of CBD [19]. Authors indicate that CBD effectively eradicates viral RNA expression in human cells by preventing protein translation and other cellular alterations. Surprisingly, unlike in other studies [5], CBD did not inhibit the activity of M^pro^ protease [19]. Authors suggested that suppression of the infection and degradation of viral RNA caused by CBD could be explained by activation of the interferon signalling pathway. As SARS-CoV-2 primarily suppresses the pathway, which may lead to excessive release of pro-inflammatory cytokines, and therefore a cytokine storm, CBD plays a vital role by reversing the process [19].

Subsequently, authors of the study further examined the capacity of cannabinoids to prevent SARS-CoV-2 infection. Analysis of over 93,000 patients at the University of Chicago Medical Center tested whether the usage of cannabinoids would result in a different SARS-CoV-2 incidence rate compared to the non-cannabinoid group. The results showed that only 5.7% of patients with a previous record of using cannabinoids (approximately 400 patients) tested positive for SARS-CoV-2, compared to 10% in the non-cannabinoid group. Moreover, patients taking CBD had an even lower rate of positive test results—1.2%—compared to the group using other cannabinoids—7.1% [19].

The findings, collectively with the in vitro studies, suggest protective properties of CBD against SARS-CoV-2 infection. In vitro reports indicate that the inhibitory effect of CBD on viral replication may be at least as potent as remdesivir, an antiviral drug already used in treatment of COVID-19 patients [26]. The antiviral properties of phytocannabinoids are presented in Table 1. Due to the lack of sufficient knowledge about the mechanisms of antiviral action modes of cannabinoids in COVID-19 infection, and sometimes even contradictory reports, it is necessary to conduct further studies that would allow to clearly verify their effectiveness.

## 3. Anti-Inflammatory Properties of Cannabinoids

In the lung cells of patients infected with SARS-CoV-2 there were detected increased levels of C reactive protein (CRP), pro-inflammatory cytokines, such as IL-1β, IL-6, IL-7, IL-8, and IL-9, fibroblast growth factor (TGF), interferons (IFNs), granulocyte colony-stimulating factor (G-CSF), granulocyte-macrophage colony-stimulating factor (GM-CSF), tumor necrosis factor (TNF), macrophage inflammatory protein 1α (MIP-1α) and vascular endothelial growth factor (VEGF), which are associated with the severe course of the disease [5].

Evidence suggests that the severity of COVID-19 is generally associated with a cytokine storm, primarily driven by the pro-inflammatory cytokines—IL-6, IL-8 and TNF-α, the overproduction of which leads to impaired oxygen diffusion, pulmonary fibrosis and multi-organ failure [24]. In vivo studies prove the inhibitory effect of cannabinoids on the course of cytokine storms, improvement of lung tissue function and reduction in inflammation [24,29,30], which may prove beneficial in the treatment of COVID-19. The anti-inflammatory effect of cannabinoids is based both on the regulation of activity and migration of cells of the immune system (macrophages, monocytes, neutrophils, lymphocytes, dendritic cells, NK cells, fibroblasts and endothelial cells), reduction in pro-inflammatory cytokines (IL-1β, IL-2, IL-6, IL-8, IL-12, IL-17, IL-18, IFN-γ, TNF-α, monocyte chemoattractant protein 1 (MCP-1)/CCL5 and GM-CSF), as well as upregulation of the release of anti-inflammatory cytokines (IL-4, IL-10, IL-11 and TGF-β) [24].

Authors of an in vitro study investigated the effects of CB1 and CB2 agonist—WIN 55,212-2 (WIN)—in human cardiomyocytes infected with SARS-CoV-2 [31]. WIN reduced the levels of IL-6, IL-8, IL-18 and TNF-α in the infected cells, as well as attenuated cytotoxicity measured by downregulation of LDH. The findings suggest that cannabinoids, such as THC, could protect the cardiac tissue in SARS-CoV-2 infection [31].

## 4. Phytocannabinoids

CBD, due to its high effectiveness in reducing inflammation by activating CB2 receptors, may prove useful in reducing inflammation and lung damage in patients infected with SARS-CoV-2 [5,32].

An in vitro study examining the anti-inflammatory effects of CBD in alveolar and macrophage models revealed the capacity of CBD to decrease secretion of inflammatory cytokines, such as IL-6, IL-8, CCL2 and CCL7, which are involved in the cytokine storm in severe COVID-19 patients [27].

In order to determine the influence of CBD on ARDS, lung tissues of wild-type mice with induced inflammation were examined after intraperitoneal administration of CBD [33]. The effects caused by Poly(I:C) consisted of reduced blood oxygen saturation, pulmonary oedema, fibrosis and significant inflammatory infiltration. Administering CBD resulted in total or partial reversion of these symptoms and led to a decrease in IL-6 levels, as well as neutrophil and lymphocyte infiltration [33]. The mechanism responsible for the effect can be explained by the upregulation of apelin—a peptide modulating central and peripheral immunity that was significantly downregulated in cases of ARDS in mice [34].

Moreover, activation of CB2 receptors in the course of respiratory syncytial virus (RSV) infection in mice limits immune cell infiltration in the lungs, downregulates levels of IFN-γ and MIP-1α, as well as increases production of IL-10 [35]. CB1 receptors also seem to mitigate the inflammatory response in RSV infection in animal models by decreasing immune cell influx, cytokine production and alleviating lung pathology [36].

In animal models, THC also seems to mitigate the inflammatory response in the airways [37]. Interestingly, the improvement was also present in mice with blockade of CB1 and CB2 receptors, indicating the involvement of alternative paths of cannabinoids in immunomodulation [37]. Collectively, in experimentally inflamed lung tissue THC induces apoptosis of mononuclear cells infiltrating lungs, modifies the metabolism of T lymphocytes, as well as downregulates alveolar macrophages, neutrophils, lymphocytes CD4+, CD8+, NK, NKT cells and proinflammatory cytokines, such as IFN-γ, IL-1β, IL-2,or TNF-α [38,39,40,41].

The ability of cannabinoids to dilate bronchi and the anti-inflammatory properties indicate the potential of cannabinoids in treating inflammatory and obstructive airway diseases [3,42]. Cannabinoid receptors appear to improve airway obturation, suffocation, cough, along with reducing leucocyte infiltration and proinflammatory cytokine levels [3,43]. The multifaceted effects of phytocannabinoids and the endocannabinoid system on the respiratory system are presented in Table 2.

On the other hand, some reports pay attention to the adverse effects of cannabinoids on the respiratory tract function [44,45]. Intensive activation of CB1 receptors can be associated with lung injury, inflammation and fibrosis, as well as increased pro-inflammatory factors in the lungs [44]. Weakening of the immune system caused by the use of cannabinoids is a potential risk factor for pneumonia and viral infections [18].

Moreover, the most common route of recreational cannabis administration is smoking with tobacco. Studies involving marijuana smokers confirm its negative impact on airway mucosa and the function of lungs [46,47]; also, it seems especially dangerous as tobacco smoking leads to upregulation of ACE2 receptors, which are known to be the gateway for SARS-CoV-2 virus to enter the host’s cells [48].

Although cannabinoids undoubtedly affect immunomodulatory responses in the respiratory system, showing significant anti-inflammatory properties, the current conclusions are majorly drawn from preclinical studies. There is still not enough data concerning the optimal dosage and treatment regimens of CBD and THC [18]. There is a pressing need for preclinical and clinical trials involving the influence of cannabinoids in SARS-CoV-2 infection. Limitations related to conducting clinical trials and latter potential implementation to medical practise include well-known psychotropic effects of cannabis, special research approval requirements in certain countries and limited supply of cannabinoids [49].

Moreover, the effect of the cannabinoids’ interactions with currently used drugs in the treatment of SARS-CoV-2 patients, such as antivirals, corticosteroids and interleukin inhibitors, remain unknown. The lack of information concerning dosing and proper routes of administration are another limitations for conducting successful future clinical trials [49]. Although, the available data is based majorly on preclinical studies in animal models and marijuana smokers, it does provide promising results.

## 5. Terpenes

Terpenes, natural compounds primarily extracted from plants, could also prove useful against SARS-CoV-2, as they are reported to exhibit anti-inflammatory, analgesic, antimicrobial and antiviral properties [50]. The terpene profiles do not only resemble the characteristics of cannabinoids but also cause the entourage effect with cannabis compounds, and therefore could enhance their therapeutical efficacy [10]. An in vitro study examined the effect of using a formulation consisting of 30 natural terpenes found in cannabis—mainly β-caryophyllene (BCP), eucalyptol and citral [50]. The antiviral activity of the formulation was enhanced when applied together with CBD, suggesting either a synergetic or additive effect between the terpene formulation and CBD.

Of all terpenes, it is BCP that seems to present the most promising therapeutical potential. BCP, a naturally occurring cannabinoid ligand, binds to various receptors, such as CB2, PPAR-α, PPAR-γ, opioid, histaminergic, TRPV and toll-like receptors (TLR) [6]. BCP was shown to target SARS-CoV-2 M^pro^ [51], which may prevent viral replication in the host’s cells. Moreover, BCP seems to target immune genes, involved in regulating numerous signalling pathways, and therefore was suggested useful as a possible agent against COVID-19 [6]. It could also be considered as a potential therapeutic candidate in COVID-19 due to its antiviral and anti-inflammatory properties, which may result in inhibition of the cytokine storm in COVID-19 patients.

CB2 receptor activation has been indicated as a potential therapeutic target in SARS-CoV-2 infection due to its anti-inflammatory activity, based on mediating several pathways involved in inflammatory response, such as inhibition of pro-inflammatory cytokines and chemokines, together with the inhibition of macrophage infiltration [6]. As a CB2 agonist, BCP may potentially limit the severity of COVID-19 by attenuating inflammation, oxidative stress, apoptosis, fibrosis and immune modulation [6].

PPAR-γ plays a role in inhibiting the replication of several viruses, such as human immunodeficiency virus (HIV), respiratory syncytial virus (RSV), hepatitis B (HBV) and hepatitis C (HCV) viruses. The activation of PPAR-γ in alveolar macrophages was shown to effectively mitigate pulmonary inflammation and reduce the excessive production of proinflammatory cytokines [6,52]. Therefore, PPAR-γ agonists, such as BCP and CBD, could prove effective in treatment of inflammatory and viral diseases [6,25].

## 6. Anxiolytic Properties of Cannabinoids

Endocannabinoid signalling is essential for stress adaptation and can be downregulated in the presence of external stressors causing anxiety and depression, such as domestic burdens, violence or increased workload [53]. The COVID-19 pandemic has led to a worldwide crisis stemming from both the fear of lockdown and the disease itself. Therefore, resilience and appropriate coping strategies become necessary in order to reduce the negative influence and prevent chronic stress.

CBD, a compound with anxiolytic, antidepressant and antipsychotic properties, could prove beneficial in decreasing stress and isolation-induced aggression, as well as help to restore the endocannabinoid homeostasis [53]. Small randomised controlled clinical trials and observational studies proved the potential of CBD in reducing anxiety among various groups of patients [54]. Observational and preclinical studies also support CBD’s therapeutic efficacy in improving the quality of sleep and reducing depression, which are often associated with anxiety [54].

The use of CBD alone, or in combination with THC or terpenes, as an adjuvant therapy, could improve the quality of life of patients with SARS-CoV-2 infection and mitigate the stress symptoms that may develop after recovery [55]. Although clinical trials are in their infancy, cannabinoids seem to limit the severity of COVID-19 based on the existing preclinical studies [55]. However, before the routine use of cannabinoids could be implemented, more evidence is needed.

## 7. Summary

The emergence of new variants of the SARS-CoV-2 virus, and the secondary states of the COVID-19 pandemic, undoubtedly caused many limitations in the daily functioning, treatment and research. On the other hand, all efforts are focused on investigating the virus, the search for vaccinations and, above all, substances and drugs that could potentially be used in both prophylaxis and treatment. Our knowledge of the anti-inflammatory and antiviral properties of cannabis have contributed to the complex analysis of the plant’s potential, particularly primary cannabinoids (CBD), secondary cannabinoids (CBG) and terpenes. The multifaceted anti-inflammatory modes of action, especially if we take into account the properties of the whole plant, could provide a more complete effect and become a practical choice in everyday clinical practice in the near future. However, assessing the properties of cannabis is significantly more difficult than studying its individual components and there is still a pressing need for more clinical trials, which would help to determine the full potential of this plant both in case of COVID-19 and many other disorders.

## Figures and Tables

**Figure 1 ijms-23-04170-f001:**
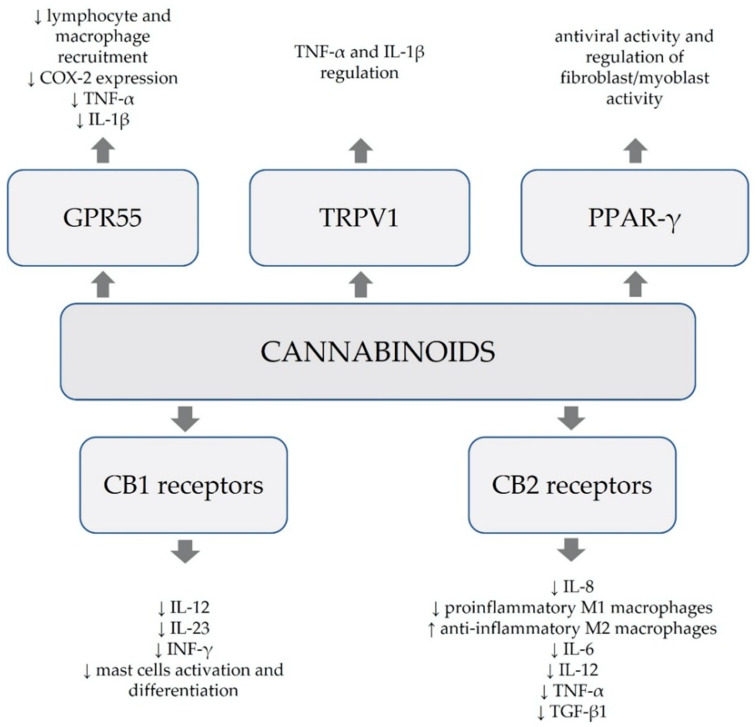
Anti-inflammatory, immunological and antioxidative modes of action of cannabinoids [3].

**Table 1 ijms-23-04170-t001:** The antiviral properties of phytocannabinoids.

CBD, THC	Inhibiting the virus translation and replication by supressing SARS-CoV-2 M^pro^ protease.	[5]
CBD	Preventing entry into host cells by inhibition of TMPRSS2.	[17,25]
High-CBD cannabis extract, CBD	Reducing susceptibility to infection by downregulation of the expression of ACE2 receptors.	[17,27]
CBD	Eradicating viral RNA expression in cells by preventing protein translation and other cellular alterations—potentially through activation of the interferon signalling pathway.	[19]
CBGA, CBDA	Preventing the SARS-CoV-2 infection at the point of cell entry.	[28]

CBD, cannabidiol; THC, Δ9-tetrahydrocannabinol; TMPRSS2, transmembrane serine protease 2; ACE2, angiotensin converting enzyme 2; CBGA, cannabigerolic acid; CBDA, cannabidiolic acid.

**Table 2 ijms-23-04170-t002:** The multifaceted effects of phytocannabinoids and the endocannabinoid system on the respiratory system.

Cannabinoids (generally)	Inhibition of cytokine storms, improvement of lung function, reduction of inflammation, bronchi dilatation, alleviation of symptoms (suffocation, cough).	[3,24,29,30,42]
CBD	Decrease of secretion of inflammatory cytokines, such as IL-6, IL-8, CCL2, and CCL7.	[27]
CBD	Improvement of blood oxygen saturation and the function of lung tissue; decrease in IL-6 levels, as well as neutrophil and lymphocyte infiltration.	[33]
THC	Apoptosis of mononuclear cells infiltrating lungs, downregulation of alveolar macrophages, neutrophils, lymphocytes CD4+, CD8+, NK, NKT cells and proinflammatory cytokines (IFN-γ, IL-1β, IL-2, or TNF-α).	[38,39,40,41]
CB2 receptors	Downregulation of immune cell infiltration in the lungs, as well as levels of IFN-γ, MIP-1α; increased production of IL-10.	[35]
CB1 receptors	Reduction of immune cell influx, cytokine production, and alleviation of lung pathology.	[36]

CBD, cannabidiol; CCL, CC chemokine ligand; THC, Δ9-tetrahydrocannabinol; NK, natural killer; NKT, natural killer T; CB, cannabinoid; IFN-γ, interferon γ; MIP−1α, macrophage inflammatory protein 1α.

## Data Availability

Not applicable.

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
