# Peer review of "Anti-Inflammatory and Antiviral Effects of Cannabinoids in Inhibiting and Preventing SARS-CoV-2 Infection"

_ijms, 2022, doi:10.3390/ijms23084170_

Round 1
Reviewer 1 Report
Congratulations for your article; What concerns us is a review of different treatments associated with the CB1 and CB2 cannabinoid receptors and how their properties can be beneficial for the treatment and prevention of SARS-CoV-2 as well as its symptoms.
I consider that it is an article that should be published because of the content and because of the continent.
All the concepts that they expose are correctly referenced and substantiated with citations. When an acronym or abbreviation appears, the full name is previously defined.
It deserves to be published from my point of view.
Author Response
Thank you very much for such favourable evaluation of our paper. We were very pleased do read your review.
I enclose the latest version of the manuscript, after a few minor alterations.

Reviewer 2 Report
This manuscript provides an overview on the potential use of cannabinoids from Cannabis sativa for the mitigation or treatment of COVID-19. Cannabinoids have both anti-inflammatory and antiviral activities, which seem to be perfect for the treatment of COVID-19. Some patients with COVID-19 also have severe lung inflammation, therefore, the dual effect, both anti-inflammatory and antiviral properties, of cannabinoids can provide the benefits for the reduction of COVID-19 symptoms. This review is well written, and the information is useful for readers in the field. This manuscript is highly recommended for publication after minor revision. In order to improve this manuscript, please consider the comments and suggestions which are listed below.
- “These difficult circumstances have also spurred the re-examination of the potential of natural substances contained in Cannabis Sativa L.”; the correct scientific name is “Cannabis sativa L.”. Please correct this typographical error.
- “It is worth noting that the anti-inflammatory and antiviral potential is shown not only by well-known cannabinoids such as cannabidiol (CBD)”; please use “….the anti-inflammatory and antiviral potentials are shown not only by…”.
- Keywords should include the word “Cannabis sativa”. Please also revise “terpens” to “terpene”.
- “This article presents a narrative review of the current knowledge in this area available in PubMed, Scopus and Web of Science databases.”; this sentence appears twice both in the abstract and in the content (lines 104-106). Please revise one of them; it should not be repeated.
- Compounds with both anti-inflammatory and antiviral properties are of great interest for COVID-19, for example, a peptide A-3302-B, please see The Peptide A-3302-B Isolated from a Marine Bacterium Micromonospora sp. Inhibits HSV-2 Infection by Preventing the Viral Egress from Host Cells. Int. J. Mol. Sci. 2022, 23, 947. https://doi.org/10.3390/ijms23020947. It is worth mentioning on this point.
- “which later become cleaved by SARS-CoV-2 proteases—Mpro and PLpro.”; this is the first mention for the abbreviations of Mpro and PLpro. Please provide full names of Mpro and PLpro.
- “Our knowledge of the anti-inflammatory and antiviral properties of cannabis has contributed to the complex analysis…”; please use “have”.
Author Response
Thank you very much for the review. We were very pleased to read such favourable evaluation.
I believe I have addressed all of your comments (7). I have found them very perceptive and helpful.
"Cannabis Sativa L." was changed to “Cannabis sativa L.” - typographical error. (Line 18)
“It is worth noting that the anti-inflammatory and antiviral potential is shown not only by well-known cannabinoids such as cannabidiol (CBD)” was changed to “….the anti-inflammatory and antiviral potentials are shown not only by…”. (Line 27)
As the reviewer kindy suggested, we changed two keywords: "cannabis" to “Cannabis sativa” and “terpens” to “terpene”. (Lines 33-34)
As the sentence “This article presents a narrative review of the current knowledge in this area available in PubMed, Scopus and Web of Science databases.” appears twice both in the abstract and in the content (line 105) we modified the second one so that it would not be repeated.
We mentioned the suggested article (https://doi.org/10.3390/ijms23020947) as its content is relevant for the manuscript and it will benefit from this alteration. (Lines 126-128)
We provided full names of Mpro and PLpro proteases as they were mentioned for the first time in the line 121.
We used "have" in the sentence “Our knowledge of the anti-inflammatory and antiviral properties of cannabis has contributed to the complex analysis…” instead of "has". (Line 340)
